# Experimentally realized physical-model-based frugal wave control in metasurface-programmable complex media

Jérôme Sol[1], Hugo Prod'homme[1], Luc Le Magoarou[1] & Philipp del Hougne [1] ✉

Metasurface-programmable radio environments are considered a key ingredient of next-generation wireless networks. Yet, identifying a metasurface configuration that yields a desired wireless functionality in an unknown complex environment was so far only achieved with closed-loop iterative feedback schemes. Here, we introduce open-loop wave control in metasurface-programmable complex media by estimating the parameters of a compact physics-based forward model. Our experiments demonstrate orders-of-magnitude advantages over deep-learning-based digital-twin benchmarks in terms of accuracy, compactness and required calibration examples. Strikingly, our parameter estimation also works without phase information and without providing measurements for all considered scattering coefficients. These unique generalization capabilities of our pure-physics model unlock unforeseen and previously inaccessible frugal wave control protocols that significantly alleviate the measurement complexity. For instance, we achieve coherent wave control (focusing or perfect absorption) and phase-shift-keying backscatter communications in metasurface-programmable complex media with intensity-only measurements. Our approach is also directly relevant to dynamic metasurface antennas, microwave-based signal processors and emerging in situ reconfigurable nanophotonic, optical and room-acoustical systems.

The tailoring of wave-matter interactions underpins the ability of wave engineers to mold the flow of information in applications spanning from communications via sensing to computing. Traditional approaches rely on fabricating judiciously designed scattering structures or coherently shaping input wavefronts. Recently, a new trend of tuning complex scattering media in situ with a large number of adjustable degrees of freedom (DOFs) emerges across scales and wave phenomena (see also Supplementary Note 1). Currently the most prominent example are metasurface-programmable smart radio environments, foreseen to play a pivotal role in next-generation wireless networks[1–3]. The same concept also underlies promising implementations of reconfigurable holographic antennas[4–6] and wave-based signal processors[7,8] in the microwave regime, and emerges in nanophotonics[9–11], optics[12–14] and

room acoustics[15,16], too. However, the inverse-design problem of identifying a configuration of the DOFs that yields a desired system transfer function is notoriously difficult for two reasons. First, the mapping from a configuration to its transfer function is a priori unknown in complex media for which detailed geometrical and/or material descriptions are usually not available. Second, this mapping is difficult to characterize because it is non-linear: multi-bounce paths create short-range and long-range correlations between the DOFs[17]. The lack of accurate setting-specific forward models so far precludes open-loop wave control in massively programmable complex media (MPCM), i.e., the possibility to optimize the metasurface configuration for a desired wave control functionality without additional measurements (see also Supplementary Note 2). Therefore, the potential of MPCMs is currently

[1]Univ Rennes, INSA Rennes, CNRS, IETR - UMR 6164, F-35000 Rennes, France. ✉e-mail: philipp.del-hougne@univ-rennes.fr

severely limited: to date, MPCMs are either merely deployed in random configurations[4,5], or optimized configurations are identified in prohibitively long closed-loop iterative in situ optimizations requiring measurements at each iteration[3,6–9,11,12,14–16]. Meanwhile, in the signal-processing community, algorithmic developments largely resort to simplified linear cascaded models[18] that risk being incompatible with the experimental reality since they ignore these correlations[17].

Nowadays, a tempting solution to enable open-loop wave control in MPCMs is to blindly combine massive amounts of data and computing power to obtain a deep-learning digital twin of an MPCM[19]. However, as we show below, such approaches surprisingly struggle to accurately learn the DOFs' correlations. Instead, by noting that the fundamental wave-physical principles underlying an unknown MPCM are easily formulated, we here report on the successful calibration and use of a compact and highly accurate physical model in a prototypical experiment with a metasurface-programmable chaotic cavity. Within the context of smart radio environments, our work experimentally achieves accurate end-to-end channel estimation, even under rich scattering conditions, overcoming the inherent limitations of existing channel-estimation algorithms[20–22] originating from their tacit linearity assumption. Beyond its accuracy and compactness, the appeal of our physical model lies in its unique far-reaching generalization capabilities that enable the retrieval of information about phases and scattering coefficients that were not included in the calibration data, enabling, for instance, non-coherent channel estimation. Thereby, we unlock previously unimagined wave control regimes for MPCMs that are inaccessible with closed-loop approaches or deep-learning digital twins.

The physics of MPCMs fundamentally differs from traditional approaches to controlling wave-matter interactions (see Supplementary Note 1). Generally speaking, the transfer function of a linear system is related to the inverse of the system's interaction matrix. The diagonal [resp. off-diagonal] entries of the interaction matrix capture the local [resp. non-local] properties of the primary entities that make up the system, i.e., their polarizabilities [resp. coupling] which depend [resp. do not depend] on the scattering occurring away from these entities (see Supplementary Note 3C for details). In wavefront shaping, the system transfer function is static, and the output linearly depends on the input which is optimized[23]. For a static inverse-designed structure, the entire interaction matrix is optimizable in the offline design phase (within physical bounds and fabrication constraints) to achieve a desired transfer function. While blind neural surrogate forward models have successfully mapped design DOFs to the corresponding system transfer functions[24–27], there are also various efforts to integrate some physics knowledge into such neural models[28–31]. In contrast, for an MPCM, only some local scattering properties (i.e., some diagonal entries of the interaction matrix) are tunable in situ[32]. A few models capturing this fact based on discrete-dipole approximations[32–34] or impedance matrices[35–37] were recently proposed but have to date not been validated experimentally, let alone in an unknown complex medium. The favorable inductive bias of physics-based models for end-to-end metasurface-parametrized channel estimation has to date not been recognized. Moreover, the naturally built-in constraints of physics-based models imply powerful generalization capabilities that have gone unnoticed, enabling unforeseen frugal wave control protocols for MPCMs.

In this article, we introduce and experimentally demonstrate physical-model-based protocols for frugal coherent wave control in metasurface-programmable complex media. The key ingredients are (i) a compact physical model whose number of parameters does not depend on the environment's complexity and that does not require an explicit description of the unknown environment, and (ii) the estimation of the physical model's parameters. Compared to deep-learning benchmarks, our physical model offers orders-of-magnitude improvements in terms of compactness (the number of model parameters), accuracy, and the number of required calibration examples. Moreover,

our approach enjoys a favorable scaling of the number of required calibration examples with the number of considered scattering coefficients. Strikingly, we discover that natural constraints built into the physical model enable the complete and accurate estimation of its parameters without every measuring phase, or without ever measuring some of the considered scattering coefficients. Thereby, we unlock previously unimagined frugal wave-control paradigms. For instance, we demonstrate how to tune a reconfigurable complex medium to feature a coherent perfect absorption (CPA) state at a desired frequency and how to identify the corresponding CPA wavefront – without ever measuring phase. Within the context of smart radio environments, our work reports the first experimental validation of a physics-compliant end-to-end channel model, the first exploration and demonstration of physics-compliant end-to-end channel estimation, and the discovery of previously unimagined frugal channel-estimation procedures (e.g., with non-coherent detection) enabled by the physics-compliant model. Our results reveal that the higher mathematical complexity of physics-compliant channel models in contrast to widespread cascaded channel models yields unique advantages for end-to-end channel estimation.

## Results
### Physical model
Our experimental system, a metasurface-programmable chaotic cavity connected via $N_A = 4$ antennas to the outside world, is shown in Fig. 1. The metasurface contains $N_S = 68$ 1-bit programmable meta-atoms whose resonance frequencies can be individually reconfigured via the bias voltages of PIN diodes (see Methods). Our system's exact geometry and material composition is unknown, and there is strong modal overlap (see Methods). However, it is easy to measure the transfer function (i.e., the $4 \times 4$ scattering matrix **S** defined through the antennas, or a subset thereof) for a few known random metasurface configurations. Is that enough to calibrate a model that accurately predicts the transfer function for any of the $2^{68}$ possible metasurface configurations?

Our system contains $N = N_A + N_S$ wireless entities of primary interest ($N_A$ antennas to input/output waves, $N_S$ meta-atoms to control the system transfer function) that are naturally discrete. We model each entity as a dipole characterized by its polarizability ($\alpha_A$ for the nominally identical antennas; $\alpha_0$ or $\alpha_1$ for the nominally identical 1-bit programmable meta-atoms, depending on their configuration). The $i$th and $j$th dipoles are coupled reciprocally via the background Green's function $G_{ij} = G_{ji}$ between their respective locations. Importantly, $G_{ij}$ accounts for all scattering in the complex environment irrespective of the latter's complexity and whether the scattering objects are continuous or discrete (see Supplementary Note 3C regarding the relation to refs. 32,37,38. concerned with discrete scattering objects surrounded by free space). The dipole moment of the $i$th dipole is proportional to the field component $E_i$ incident at its location along its orientation: $p_i = \alpha_i E_i$, where $E_i = E_i^{\text{ext}} + \sum_{j=1}^{N} G_{ij} p_j$ is the superposition of the externally incident wave $E_i^{\text{ext}}$ on the $i$th dipole and the waves re-radiated from the other $N - 1$ dipoles. Solving self-consistently for the dipole moments, we obtain (in matrix form) $\mathbf{p} = \mathbf{W}^{-1} \mathbf{E}^{\text{ext}}$, where the diagonal and off-diagonal entries of the interaction matrix **W** are $\alpha_i^{-1} - G_{ii}$ and $-G_{ij}$, respectively (see Supplementary Note 3C for details). The input and output wavefronts are proportional to the entries of $\mathbf{E}^{\text{ext}}$ and **p**, respectively, that correspond to the $N_A$ antennas. Because we seek to calibrate our model to an experimental setting, any proportionality factors and additive constants that do not depend on the metasurface configuration cannot be unambiguously identified and can hence be absorbed into $\mathbf{W}^{-1}$, meaning that we can work with $\mathbf{S} = [\hat{\mathbf{W}}^{-1}]_{\mathcal{A}\mathcal{A}}$ and $\mathbf{H} = [\hat{\mathbf{W}}^{-1}]_{\mathcal{R}\mathcal{T}}$ here, where $\mathcal{T}$ and $\mathcal{R}$ denote the dipole indices associated with transmitting and receiving antennas, respectively, and $\mathcal{A} = \mathcal{T} \cup \mathcal{R}$. We use the circumflex symbol (^) to denote that we have absorbed

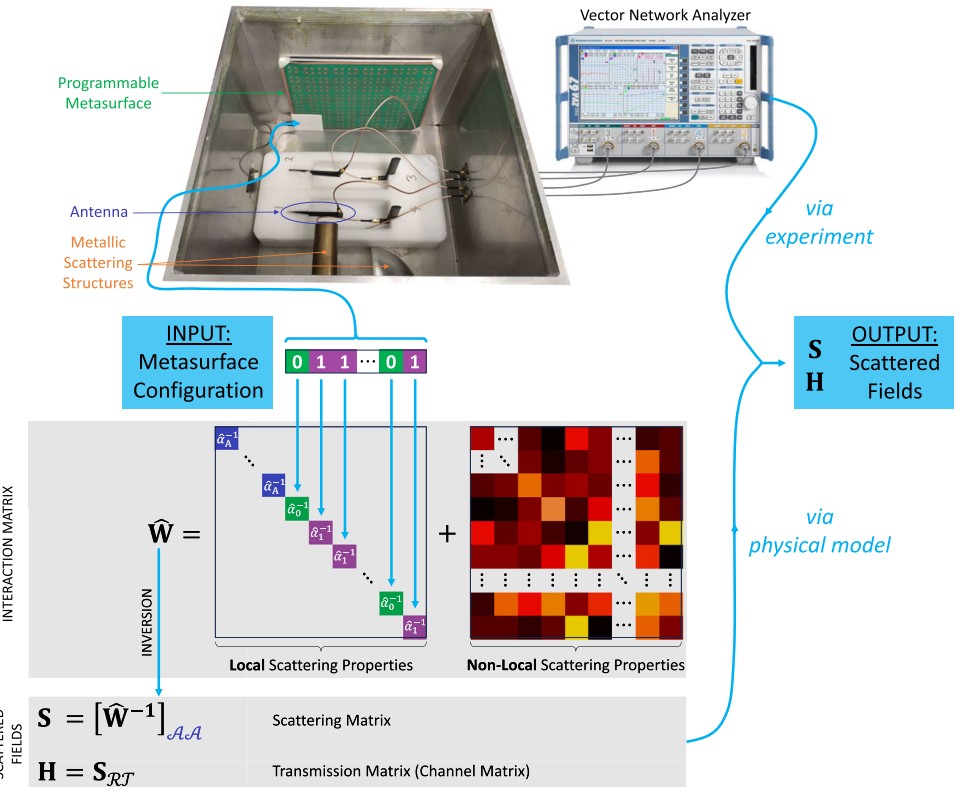

**Fig. 1 | Physical model.** The scattering of electromagnetic fields inside a complex massively programmable wave system depends in a highly non-trivial way on the configuration of a programmable metasurface due to non-local interactions between the scattering entities. The physical model faithfully reproduces the experimental scattering process. It obtains the scattered fields from a block of an inverted interaction matrix $\hat{\mathbf{W}}$ that captures the local properties of the system's primary internal scattering entities (antennas and programmable meta-atoms) as well as their non-local interactions via proximity and long-range reverberation. The physical model is calibrated with a very small random subset of all possible metasurface configurations and corresponding scattering measurements from the experimental system of interest, without any information about its geometry. In the photographic image, the top cover is removed to show the cavity's interior.

multiplicative and additive factors into the respective quantity. This compact physical model (summarized in Fig. 1) involves three complex-valued local parameters ($\hat{\alpha}_A$, $\hat{\alpha}_0$ and $\hat{\alpha}_1$) and $(N+1)N/2$ complex-valued non-local parameters ($\hat{\mathbf{W}}$ is symmetric due to reciprocity); importantly, its size is independent of the scattering environment's complexity and dimensions (2D vs 3D).

Clearly, there are infinitely many interaction matrices that yield the same transfer function, so there is no unique true set of parameters to describe a given experimental system. Rather than being a problem, this non-uniqueness makes it easier to calibrate our model. (See ref. 39 for a discussion under which conditions almost all ambiguities can be lifted.) We refrain from imposing additional constraints besides reciprocity (such as passivity) for the same reason. Note also that we could not have formulated our model based on some form of temporal coupled-mode theory[40] because the functional dependence of local and non-local properties on the metasurface configuration is unknown in the modal basis. Using an efficient gradient-descent algorithm (see Methods) and a calibration data set involving $N_{data}$ pairs of known random metasurface configurations and the corresponding measured transfer functions, we calibrate our model (i.e., estimate its parameters; see "Methods").

**Benchmarking and scaling of model accuracy**
We now investigate how the accuracy of our calibrated physical model depends on $N_{data}$ as well as the number of considered scattering coefficients, and we compare the performance to two benchmark models: first, a simple linear model, being the currently most widely used model in the signal-processing community[18], and, second, a deep-learning (DL) digital twin[19] (see "Methods"). The linear model can be derived from our physics-based model upon expressing the

underlying matrix inversion as infinite series of matrix powers and truncating that series after the first term depending on the metasurface configuration[17]. The DL model is a standard multilayer-perceptron feedforward artificial neural network that is widely used for blind function approximation without any a priori knowledge. Of course, the more valid a priori knowledge is injected into a model, the better its inductive bias will be; therefore, *in hindsight* it will be obvious that our physics-based model outperforms the benchmarks (i.e., taking its validity as established). We quantify the accuracy with a metric defined analogous to the signal-to-noise ratio: $\zeta$ is the ratio of the variance of the true scattering coefficient and the variance of the difference between the true and predicted scattering coefficient, the variance being taken over random unseen metasurface configurations[17]. To contextualize this metric, note that $\zeta = 0$ dB is trivially achieved by assuming the scattering coefficient does not depend on the metasurface configuration, and that improving $\zeta$ from 10 dB to 30 dB can yield a 2.5-fold improvement in the channel's information-transfer capacity (see Fig. S4).

To start, we examine the accuracy on unseen random metasurface configurations with the largest considered value of $N_{data} = 4 \times 10^4$ to predict a single transmission coefficient ($S_{31}$). The linear model (Fig. 2a) uses few parameters but its accuracy of $\zeta_{31} = 7.3$ dB is quite low since by construction it cannot capture multi-bounce paths encountering multiple meta-atoms. The DL model uses three orders of magnitude more parameters and manages to capture some (but surprisingly not all) of the non-linear correlations between meta-atoms, achieving $\zeta_{31} = 13.9$ dB. Visual inspection of Fig. 2b reveals the DL model's limited accuracy. In contrast, the physical model uses two orders of magnitude fewer parameters than the DL model and achieves an order of

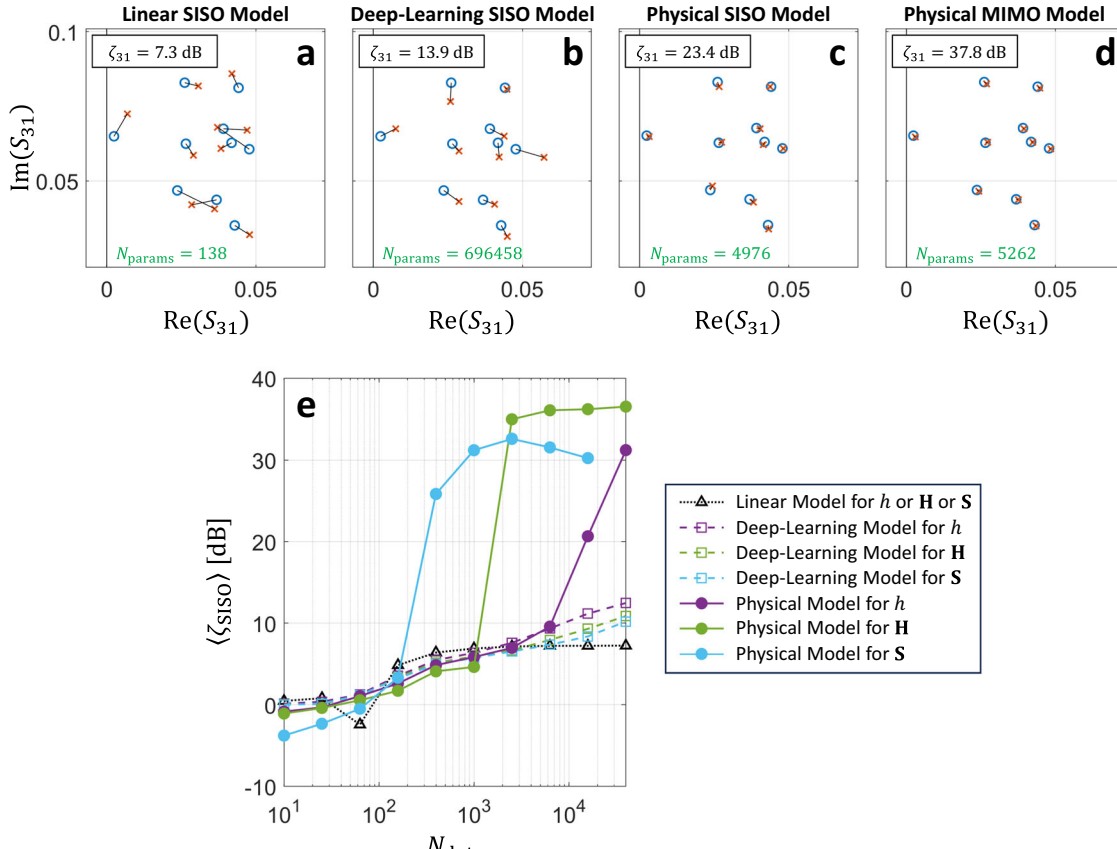

**Fig. 2 | Benchmarking and scaling of model accuracy. a–d,** Measured ground truth (blue circle) and model prediction (red cross) of $S_{31}$ for ten unseen random metasurface configurations for the linear model (**a**), the deep-learning model (**b**), and the physical model (**c**), all calibrated only with $S_{31}$, as well as for the physical model calibrated with the $2 \times 2$ transmission matrix (**d**). The achieved accuracy $\zeta_{31}$ and number of model parameters $N_{\mathrm{params}}$ are indicated. **e** Scaling of the accuracy $\zeta_{\mathrm{SISO}}$ of a single predicted transmission coefficient as a function of the size $N_{\mathrm{data}}$ of the calibration data set. We consider the cases of models calibrated to predict only a single transmission coefficient ($h$, purple), a $2 \times 2$ transmission matrix (**H**, green), and a $4 \times 4$ scattering matrix (**S**, blue). We compare these cases of the physical model and the two benchmark models (DL and linear). For the linear model, the accuracy does not depend on the number of predicted scattering coefficients. In the cases of models predicting **H** or **S**, $\zeta_{\mathrm{SISO}}$ is averaged over all available transmission coefficients.

magnitude better accuracy of $\zeta_{31} = 23.4$ dB as seen in Fig. 2c. If the physical model is calibrated for the $2 \times 2$ transmission matrix rather than only the single transmission coefficient $S_{31}$, then its accuracy improves by yet another order of magnitude without requiring significantly more parameters – see Fig. 2d. Calibrating our physical model with more scattering parameters helps because it understands how they are related. In contrast, the linear model predicts each scattering coefficient independently, and the DL model's accuracy even deteriorates slightly when asked to predict multiple scattering parameters (see Fig. 2e). In Fig. S3 we also show a supplemental experiment conducted in a meeting room; the effect of reverberation therein is weaker such that the benchmark models perform better, but the physical model still is one order of magnitude more accurate while using hundred times fewer parameters.

As $N_{\mathrm{data}}$ is increased, we observe in Fig. 2e that the linear model's accuracy eventually saturates, the DL model's accuracy slowly but steadily improves, and the physical model's accuracy slowly improves until at some value of $N_{\mathrm{data}}$ the accuracy suddenly improves by one or multiple orders of magnitude before saturating. This accuracy jump is reminiscent of phase transitions in compressive sensing[41]. The more scattering coefficients are used to calibrate our physical model, the earlier the phase transition occurs. In the case of calibrating with the full scattering matrix, $N_{\mathrm{data}} = 398$ is already sufficient to achieve $\zeta = 25.8$ dB, a value that the benchmark models never reach within the considered range of $N_{\mathrm{data}}$. (Note that $398 \ll 2^{68}$, $2^{68}$ being the total number of possible metasurface configurations).

## Phase retrieval

Since our physical model understands the wave physics at play, we now explore to what extent its generalization capabilities go beyond predicting transfer functions for unseen metasurface configurations. In this section, we calibrate our physical model with phaseless data and surprisingly find that it nonetheless accurately predicts all phase relations. Whereas neither the linear nor the DL model can make any meaningful phase prediction without calibration data involving phase information, the physical constraints built into our model seemingly imply that if it correctly predicts amplitudes then it must have simultaneously understood the phase relations.

For concreteness, we calibrate our phase-retrieval physical model (PR-model) with intensity-only information of the full $4 \times 4$ scattering matrix for $N_{\mathrm{data}} = 10^5$ random metasurface configurations. Our PR-model's complex-valued predictions for unseen random metasurface configurations are seen in Fig. 3a–d to accurately predict how the phase of each scattering coefficient depends on the metasurface configuration, as well as what the relative phase between different scattering coefficients is. The achieved $\zeta$ around 20 dB exceeds what the benchmark models achieve with complex-valued calibration data. Retrieving phase information from phaseless data is an established discipline within signal processing with applications across the EM spectrum because it removes the costly need for coherent detection[42–44]. However, conventional phase-retrieval deals with static rather than programmable systems and uses elaborate algorithms to retrieve either the system's transfer function or the input wavefront.

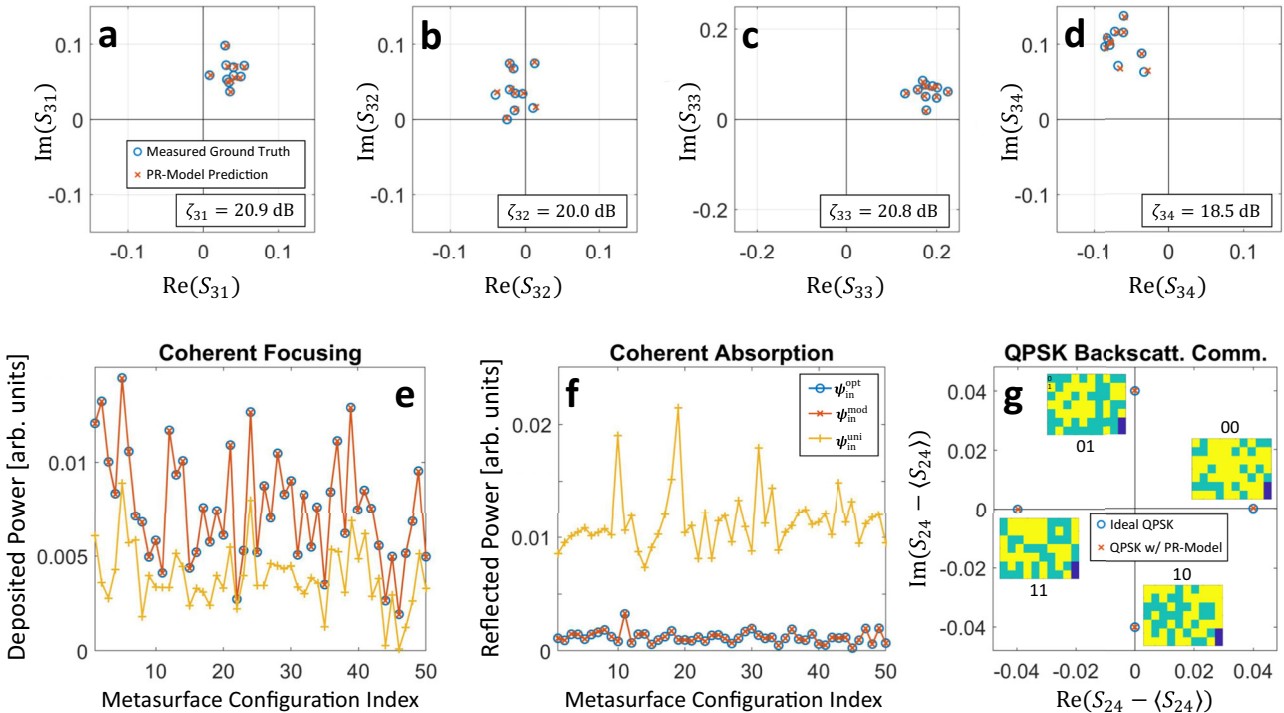

**Fig. 3 | Physical-model-based coherent wave control with phaseless calibration data. a–d** Measured ground truth and phase-retrieval physical-model (PR-Model) predictions for four selected scattering coefficients (for ten random unseen metasurface configurations). The PR-Model was calibrated for the 4 × 4. scattering matrix without any phase information. **e** Deposited energy at port 1 upon injecting a coherent wavefront through the remaining three ports (for 50 random unseen metasurface configurations). $\psi_{\text{in}}^{\text{opt}}$ (blue) is the benchmark (and provably optimal) wavefront obtained via phase conjugation given perfect knowledge of the relevant scattering coefficients, $\psi_{\text{in}}^{\text{mod}}$ (red) is obtained with the same approach but using the PR-Model, and $\psi_{\text{in}}^{\text{uni}}$ (yellow) is a uniform wavefront (see Methods for

details). **f** Reflected power upon injecting a coherent wavefront through all four ports (for 50 random unseen metasurface configurations). High absorption corresponds to low reflected power. $\psi_{\text{in}}^{\text{opt}}$ (blue) is the benchmark (and provably optimal) wavefront obtained via an eigendecomposition of $\mathbf{S}^{\dagger}\mathbf{S}$ assuming perfect knowledge of $\mathbf{S}$, $\psi_{\text{in}}^{\text{mod}}$ (red) is obtained with the same approach but using the PR-Model, and $\psi_{\text{in}}^{\text{uni}}$ (yellow) is a uniform wavefront (see "Methods" for details). **g** Identification using the PR-Model of four metasurface configurations (shown as insets) that enable quadrature-phase-shift-keying (QPSK) backscatter communications when port 2 radiates a continuous-wave signal and port 4 detects the received phase (or vice versa) (see "Methods" for details).

Our results suggest that if system programmability is available, a simple compact model and gradient descent algorithm is sufficient because wave-physical constraints naturally built into the model are leveraged.

Using our PR-model, we can perform coherent wave control at a level comparable to knowing the ground-truth scattering matrix—even though we never measured phase. Phaseless coherent wave control in MPCMs has not been imagined previously, and it was out of reach with closed-loop approaches or deep-learning forward models. First, we consider the standard wavefront-shaping problem of coherent focusing on an antenna by injecting a coherent wavefront through the remaining antennas. The provably optimal input wavefront is the phase-conjugated transmission vector. Applying phase conjugation to the predictions of our PR-model, we achieve on average 99.92 % of the ideal focusing efficiency with the ground-truth complex-valued transmission vector (see Fig. 3e). To underline the importance of phase information, we show that an arbitrary input wavefront (e.g., a uniform one) achieves only 69.81 % of the ideal focusing. Since we are able to accurately identify the ideal wavefront for any metasurface configuration, we can also combine control over the input wavefront and the metasurface configuration. By selecting the best metasurface configuration out of the $10^5$ ones used for phaseless calibration, we achieve a deposited energy of 0.0198 arb. units using our PR-model, compared to 0.0201 arb. units with ideal ground-truth knowledge. Another iconic example of coherent wave control is coherent absorption: by tailoring the input wavefront, the energy absorbed within the system can

be maximized such that as little energy as possible exits the system[45]. As in the previous example, without ever having measured phase information, we closely match the ideal performance and significantly outperform that with an arbitrary input (Fig. 3f). Again, we can also combine wavefront shaping with structural control and identify the metasurface configuration that yields the largest absorption in combination with a suitable input wavefront. Our PR-model points toward the same metasurface configuration as the ideal ground-truth knowledge, such that both achieve a minimal reflected power of −42.0 dB which is very close to zero and hence to achieving coherent perfect absorption[46] (CPA) in the system. To the best of our knowledge, no prior work proposed or experimentally achieved the tuning of a complex system to CPA at a desired frequency without ever having measured phase information.

Based on phaseless calibration data, we are also able to identify metasurface configurations that enable phase-modulated backscatter communications. In the latter, Alice encodes her message into stray ambient waves by modulating them with the metasurface such that the phase of the signal received by Bob carries Alice's message[47]. In Fig. 3g, we present the four metasurface configurations and corresponding phases measured by Bob. They closely match the ideal ones for quadrature-phase-shift-keying backscatter communications.

### Green's function retrieval
Finally, we now demonstrate that our physical model can even predict scattering coefficients for which no calibration data was ever available.

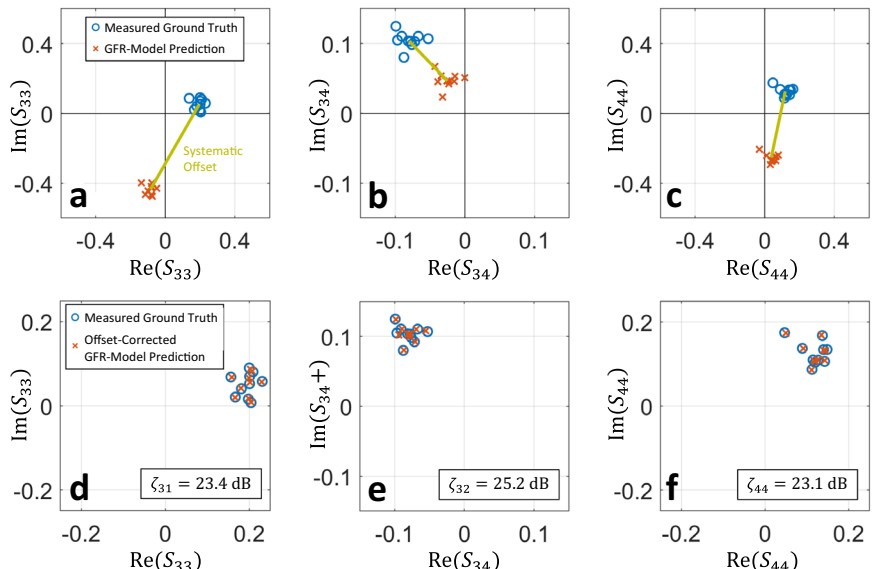

**Fig. 4 | Prediction of dependence on metasurface configuration for scattering coefficients not included in calibration data. a–c** Measured ground truth and Green's-function-retrieval physical-model (GFR-Model) predictions for three unseen scattering coefficients (for ten random unseen metasurface configurations). The GFR-Model was calibrated with transmit and receive information from ports 1 and 2, but receive-only information from ports 3 and 4, i.e., without any information about $S_{33}$, $S_{34}$, $S_{43}$, $S_{44}$. The predictions of the unseen scattering coefficients suffer from a systematic constant offset (independent of the metasurface configuration), indicated by the line connecting the centers of the ground-truth cloud and the predicted cloud. **d–f** Upon correction of the systematic offset, the physical model predicts the unseen scattering coefficients with high fidelity.

This is yet another previously unforeseen type of frugal wave control in MPCMs that is inaccessible unless a physics-based model is used. Of course, this is only possible if some information about the involved antennas is available. For concreteness, we exclude one $2 \times 2$ receive block from our $4 \times 4$ scattering matrix in the calibration data. Specifically, our physical model thereby sees information about signals received by antennas 3 and 4 that were injected via antennas 1 and 2, but not about the transmission between antennas 3 and 4, nor about the reflection coefficients of antennas 3 and 4. Clearly, the benchmark models could not make any meaningful prediction of a scattering coefficient that was excluded from the calibration data. Our Green's function retrieval physical model (GFR-model), on the other hand, successfully predicts the scattering coefficients up to a systematic offset, as seen in Fig. 4. The systematic offset must correspond to inaccurately captured paths that never bounce off any of the other primary scattering entities[17]. There is no ambiguity that could justify this systematic offset, but since these paths were only probed very indirectly in the available calibration data, our GFR-model's difficulty to capture them is understandable. In some applications such as the QPSK backscatter scheme, the static component of the scattering coefficient not affected by the metasurface does not matter anyway (see "Methods"). In any case, this systematic offset is easily corrected via a single additional calibration measurement of the unseen scattering coefficients for one known metasurface configuration. The achieved $\zeta$ values (which are independent of the systematic offset) exceed 23 dB, implying that our GFR-model predicts unseen scattering coefficients more accurately than the benchmark models predict seen scattering coefficients. Coherent wave control experiments similar to the ones from Fig. 3 are shown for the present case of Green's function retrieval in Fig. S2. Particularly noteworthy is the QPSK backscatter communication therein which is established based on the unseen scattering coefficient.

Our GFR-model's ability to retrieve all essential information about unseen Green's functions is reminiscent of algorithmically very different techniques that cross-correlate two time-domain recordings of a diffuse wave field at different receivers to recover the Green's function between these receivers[48,49]. The noise can be emulated via different realizations of disorder in a chaotic cavity[50], which has even been realized by randomly configuring a programmable metasurface[51]. These cross-correlation techniques require by construction broadband calibration data and typically only retrieve information about short (i.e., few-bounce) paths. Our GFR-model is hence distinct and complementary, being (i) a monochromatic approach and (ii) very efficient at retrieving all multi-bounce paths but struggling to retrieve the single-bounce path.

## Discussion

One might wonder why our physical model describes our experiments so well even though the utilized dipole antennas and programmable meta-atoms are not very small compared to the wavelength. The reason is that structural scattering of the antennas and meta-atoms is an indistinguishable part of the background scattering in the radio environment. The dipole size that matters is the separation of the conductors in the coaxial cables in the case of the antennas, and the size of the tunable lumped elements in the case of the programmable meta-atoms; both are electrically very small. Detailed recent discussions about the role of structural scattering for physics-compliant models of metasurface-programmable complex media can be found in refs. 52,53.

To summarize, we have introduced open-loop experimental wave control in unknown MPCMs by estimating the parameters of a highly compact and accurate physics-based model from very few calibration examples. We have discovered a previously unrecognized favorable inductive bias of physics-based channel estimation, enabling orders of magnitude higher accuracy than benchmark linear and DL models, while using hundred times fewer parameters than the latter. Moreover, the required calibration data set size decreases as the number of considered scattering coefficients increases, which is very favorable given trends toward massive MIMO systems. Most remarkably, thanks to naturally built-in constraints, our physics-based model can predict information about relative phases and/or scattering coefficients that were not seen during calibration. Thereby, our model unlocks previously unimagined frugal wave control protocols for MPCMs that are inaccessible based on closed-loop experiments or linear or DL forward

models. For instance, we tuned an MPCM to CPA at a desired frequency and identified the CPA wavefront without ever measuring phase. The discovered frugal wave control is of high technological relevance because it alleviates the hardware cost of channel estimation in metasurface-programmable radio environments by enabling operation with non-coherent detection and/or partial channel sounding. Beyond this specific application, our model enables the real-time optimization of wave devices based on MPCMs, such as reconfigurable holographic antennas and routers, across the EM spectrum.

Looking forward, on the one hand, we anticipate extensions of our physical model to scenarios involving broadband operation and dynamic radio environments. The latter will enable the inevitable[54] integration of sensing and communications in next-generation networks based on a compact and accurate physical model. On the other hand, we expect our present work to unlock the door toward formulating fundamental bounds on the transfer functions that can be achieved in an MPCM, building on current theoretical works targeting physical limits of inverse-designed static structures for which the entire interaction matrix is designable[55,56].

## Methods

### Experimental setup

**Scattering environment.** Our main experimental setup is based on the irregularly shaped and electrically large metallic cavity of dimensions $0.385\,\text{m} \times 0.422\,\text{m} \times 0.405\,\text{m}$ that is shown in Fig. 1. Two metallic objects perturb the regular shape of the underlying metallic box in order to break its symmetries: a long cylinder piercing into the volume of the cavity, as well as a sphere octant placed inside the cavity. Four monopole WiFi antennas (ANT-W63WS2-ccc) are placed therein in two pairs of two parallel antennas separated by 9 cm, and the orientation of the two pairs is mutually orthogonal. These four antennas are connected to monomodal coaxial cables, such that the $4 \times 4$ scattering matrix **S** of our system is defined in an obvious manner. A four-port vector network analyzer (Rhode & Schwarz ZVA 67) is used to measure **S** (settings: 15 dBm emitted power, 1 kHz intermediate-frequency bandwidth). A programmable metasurface (detailed in the next subsection) covers parts of the cavity walls.

Based on the decay rate of the inverse Fourier transform of transmission spectra measured between different pairs of ports and for different random metasurface configurations, we estimate the system's composite quality factor as $Q = 620$[57]. Based on Weyl's law, we find that around $\mathcal{N} \sim \frac{8\pi V}{c^3 Q} f_0^3 = \frac{8\pi}{Q} \frac{V}{\lambda^3} = 14$ modes overlap at a given frequency within the considered interval, implying that we are operating in the multi-resonance transport regime.

Our supplemental experimental setup is based on the meeting room of dimensions $6.57\,\text{m} \times 3.18\,\text{m} \times 2.49\,\text{m}$ that is shown in Fig. S3a. Two horn antennas (ETS-3115) are placed therein on opposite sides of the room, facing the metasurface. These two antennas are connected to monomodal coaxial cables, such that the $2 \times 2$ scattering matrix **S** of our supplemental system is defined in an obvious manner. We measure **S** using a vector network analyzer (Keysight M9005A chassis with M9374A modules) (settings: 15 dBm emitted power, 1 kHz intermediate-frequency bandwidth). We estimate the meeting room's composite quality factor as $Q = 342$ and we find that around $\mathcal{N} \sim 2.2 \times 10^3$ modes overlap at a given frequency.

**Programmable metasurface.** The programmable metasurface prototype (purchased from Greenerwave) consists of 68 electrically thin meta-atoms (ignoring the 8 broken meta-atoms). The spacing between neighboring meta-atoms is on the order of half a wavelength (see Fig. S1b). Each meta-atom offers independent 1-bit control over the two independent field polarizations. The metasurface design follows that outlined in ref. 58 which is based on the coupling between a fixed resonator and a resonator whose resonance is controlled via the bias voltage of a PIN diode. Under normal

incidence, the two possible configurations of a meta-atom roughly mimic Dirichlet and Neuman boundary conditions in the vicinity of our working frequency of 5.2 GHz, as seen in Fig. S1. Throughout this work, we only use the metasurface's control over one polarization, and we keep its configuration for the other polarization fixed in its "0"-state throughout. The methods for estimating the parameters of our physics-based model presented in our work apply to any linear programmable metasurface, irrespective of the design, the arrangement, and the number of its programmable meta-atoms (see Supplementary Note 4).

### Modeling

**Physical-model calibration.** Our physical model (see Fig. 1) contains $N_{\text{params}} = 2(3 + (N+1)N/2)$ parameters, where $N = N_A + N_S$ is the number of primary scattering entities (utilized antennas and meta-atoms). The first and second term in this expression account for the local and non-local scattering properties (see Fig. 1). The second term accounts for the symmetry of **W** due to reciprocity. The prefactor of two accounts for the fact that complex-valued parameters have separate real and imaginary parts.

In Fig. 2, we calibrate our model for three cases: (i) a single transmission coefficient ($N_T = N_R = 1$, $N_A = 2$), (ii) a $2 \times 2$ transmission matrix ($N_T = N_R = 2$, $N_A = 4$), and (iii) a $4 \times 4$ scattering matrix ($N_A = N_T = N_R = 4$). Let $\mathcal{H}(\mathbf{c}) \in \mathbb{C}^{N_R \times N_T}$ be the ground-truth experimentally measured scalar or matrix of interest for metasurface configuration $\mathbf{c}$, and $\hat{\mathcal{H}}(\mathbf{c})$ denotes our model's prediction thereof. We use the TensorFlow library to calibrate via gradient descent with an error back-propagation algorithm our physical model's $N_{\text{params}}$ parameters given a calibration data set comprising $N_{\text{data}}$ pairs of random metasurface configurations $\mathbf{c}$ and the corresponding experimental measurements of $\mathcal{H}(\mathbf{c})$. We define our cost function to be minimized as $\mathcal{C} = \min_\theta \langle |\mathbf{y} - e^{i\theta}\hat{\mathbf{y}}| \rangle$, where $\mathbf{y} = \mathcal{H}(\mathbf{c})\mathbf{x}$ and $\hat{\mathbf{y}} = \hat{\mathcal{H}}(\mathbf{c})\mathbf{x}$ are the true and predicted vectors of measured signals, respectively, upon coherent injection of a pilot signal $\mathbf{x} \in \mathbb{C}^{N_T \times 1}$ into the system with metasurface configuration $\mathbf{c}$. We use $N_T$ pilot signals drawn from a complex-valued random distribution with normally distributed real and imaginary parts, and the pilot signals are the same for all metasurface configurations in the calibration data set. Future work can refine the utilized pilot signals. The minimization of $\theta$ in the definition of the cost function ensures that no effort is spent on learning a global phase constant without physical meaning. The averaging in the cost function is over the different pilot signals, the different metasurface configurations contained in the batch of calibration data used to evaluate $\mathcal{C}$, and the different scattering coefficients contained in $\mathcal{H}$. All model parameters are initialized randomly with values from a truncated normal distribution (mean: 0; standard deviation: 0.2). This initialization method can be refined in future work. We use a batch size of 1000 and the Adam method for stochastic optimization with an initial step size of $10^{-2}$ that is gradually reduced over the course of the optimization. We use 8/9th of the available calibration data for training and the remainder for validation. Training is stopped when the last 7500 iterations did not yield any further improvement, and the parameter settings corresponding to the best validation cost function value are restored.

To invert the interaction matrix, for each slice (i.e., each metasurface configuration) the indices of the primary scattering entities are rearranged so that entities with the same polarizability are grouped together. This yields a $3 \times 3$ block matrix (the blocks being "A", "0", and "1") to which the block matrix inversion lemma is applied multiple times until $\mathbf{S} = [\mathbf{W}^{-1}]_{AA}$ is obtained. This multi-step procedure is computationally more efficient because we only partially evaluate $\mathbf{W}^{-1}$ and, more importantly, it avoids problems that arise whenever **W** is ill-conditioned (e.g., whenever the magnitudes of $\alpha_A$, $\alpha_0$ and $\alpha_1$ are significantly different).

In Fig. 3, we follow the same procedure as in Fig. 2 except for using the modified cost function $\mathcal{C}_{\text{PR}} = \langle ||\mathbf{y}| - |\hat{\mathbf{y}}|| \rangle$ and a batch size of 10000.

Whereas for Fig. 2 the same results could have been obtained using pilot signals in the canonical basis (e.g., [0 1] and [1 0] for $N_T = 2$), in the phaseless case in Fig. 3 the relative phase relations between scattering coefficients can only be retrieved if the pilot signals are chosen such that non-zero wave amplitudes are coherently injected through multiple antennas simultaneously. This condition is trivially satisfied with random pilot signals.

In Fig. 4, we follow the same procedure as in Fig. 2 except for using the modified cost function $\mathcal{C}_{\mathrm{GFR}} = \min_{\theta}\langle|\mathbf{A} \bullet \mathcal{H} - e^{i\theta}\mathbf{A} \bullet \widehat{\mathcal{H}}|\rangle$, where $\bullet$ denotes element-wise multiplication and $\mathbf{A}$ is a $N_R \times N_T$ matrix with Boolean elements that indicate if a specific scattering coefficient is included or excluded from the calibration data set.

**Linear model.** The simplest model of how a scattering coefficient $S_{ij}$ depends on the metasurface configuration $\mathbf{c}$ assumes a linear relation between the two: $\hat{S}_{ij} = S_{ij}^0 + \boldsymbol{\tau}_{ij}^T\mathbf{c}$. (To be precise, this relation is affine rather than linear due to the non-zero constant term $S_{ij}^0$, but for simplicity we refer to it as linear throughout this paper.) $\hat{S}_{ij}$ denotes the approximation of $S_{ij}$ by the model. Within the signal-processing literature, the cascaded model $\hat{S}_{ij} = S_{ij}^{\mathrm{TX-RX}} + \mathbf{H}_{ij}^{\mathrm{MS-RX}\,T}\mathrm{diag}(\mathbf{c})\mathbf{H}_{ij}^{\mathrm{TX-MS}}$ which postulates that transmitter (TX) and receiver (RX) are linked via one direct path and one path that bounces off the metasurface (MS) once is widespread. This cascaded model can be collapsed to the above-cited linear model because in our case without any knowledge of the setup's geometry there is no unambiguous distinction between $\mathbf{H}_{ij}^{\mathrm{MS-RX}}$ and $\mathbf{H}_{ij}^{\mathrm{TX-MS}}$. A linear model clearly neglects any structural non-linearity that arises from paths that bounce off multiple meta-atoms due to proximity-induced mutual coupling or reverberation[17].

The linear model has $N_{\mathrm{params}} = 2(N_S + 1)$ parameters (separating each complex-valued parameter into real and imaginary part) per scattering coefficient. Given a calibration data set comprising $N_{\mathrm{data}}$ pairs of random metasurface configurations and the corresponding measurements of the scattering coefficient, we calibrate these $N_{\mathrm{params}}$ parameters using the linear regression function from the sklearn library in Python. Real and imaginary parts are predicted independently by the linear model.

The linear model also predicts each scattering coefficient independently. When multiple scattering coefficients are to be modeled, e.g., a transmission or scattering matrix, then the above procedure is independently performed for each scattering coefficient.

**Deep-learning benchmark.** Our deep-learning model is a standard multilayer-perceptron feedforward artificial neural network that is widely used for blind function approximation without any a priori knowledge (see ref. 19 in the MPCM context). It consists of $n = 5$ fully connected layers, each made up of $6N_S$ neurons with bias terms and non-linear ReLU activation, and an output layer with bias terms but without activation. The number of parameters is $N_{\mathrm{params}} = (N_S M + M) + (n - 1)(MM + M) + 2(MN_{\mathrm{coeff}} + 1)$, where $M = 6N_S$ and $N_{\mathrm{coeff}}$ is the number of scattering coefficients to be learned. The cost function (loss) is $\mathcal{C} = \min_{\theta}\langle|\mathcal{H} - e^{i\theta}\widehat{\mathcal{H}}|\rangle$ and we use the Adam optimizer with its default learning rate of $10^{-3}$ and a batch size of 10. Three quarters of the calibration data are used for training, and the remainder for validation. We train until the validation loss has not improved for three epochs, and we restore the weights that corresponded to the best validation loss. We found that the results are not very sensitive to the exact choice of hyperparameters. Note that by construction this deep-learning model cannot converge to the physics-based model because of its feedforward nature in contrast to the recurrent scattering occurring in the physical setup that is encoded in the matrix inversion of the physics-compliant model. Of course, this insight is only possible *in hindsight* given the results of our present work which establish the validity of the physics-based model.

## Wave-control experiments

**Coherent focusing.** We consider the problem of focusing wave energy on one of the four antennas by injecting a coherent monochromatic wavefront $\boldsymbol{\psi}_{\mathrm{in}}$ at 5.2 GHz through the remaining three antennas. The metasurface is in a known configuration $\mathbf{c}$ that is chosen randomly and has not been part of the calibration data set. $\boldsymbol{\psi}_{\mathrm{in}}$ is normalized so that its 2-norm is unity. Let $\mathbf{t}$ denote the transmission vector from the three input ports to the output port. The wave energy deposited at the output port is $\mathbf{t}^T\boldsymbol{\psi}_{\mathrm{in}}$. The provably optimal input wavefront for coherent focusing is $\boldsymbol{\psi}_{\mathrm{in}}^{\mathrm{opt}} = \mathbf{t}^*/||\mathbf{t}^*||_2$ and constitutes our benchmark (blue) in Fig. 3e and Fig. S2a. This benchmark is known as phase conjugation (or monochromatic time reversal) in wave physics and as maximum-ratio transmission in signal processing. Because of the rich scattering in our setup, this is *not* an instance of beam-forming. For physical-model-based wave control, we do not have access to $\mathbf{t}$ but only to our model's prediction $\mathbf{t}_{\mathrm{mod}}(\mathbf{c})$ thereof such that we use $\boldsymbol{\psi}_{\mathrm{in}}^{\mathrm{mod}} = \mathbf{t}_{\mathrm{mod}}^*(\mathbf{c})/||\mathbf{t}_{\mathrm{mod}}^*(\mathbf{c})||_2$ as coherent input wavefront (red). To illustrate the importance of wavefront shaping, we also show a benchmark with a uniform wavefront $\boldsymbol{\psi}_{\mathrm{in}}^{\mathrm{uni}} = [1\,1\,1]/\sqrt{3}$ (yellow). Figure 3e and Fig. S2a show the deposited energy upon injecting $\boldsymbol{\psi}_{\mathrm{in}}^{\mathrm{opt}}$ (blue), $\boldsymbol{\psi}_{\mathrm{in}}^{\mathrm{mod}}$ (red) and $\boldsymbol{\psi}_{\mathrm{in}}^{\mathrm{uni}}$ (yellow) for 50 random and previously unseen metasurface configurations. Note that the fact that the deposited energy is almost identical for $\boldsymbol{\psi}_{\mathrm{in}}^{\mathrm{opt}}$ and $\boldsymbol{\psi}_{\mathrm{in}}^{\mathrm{mod}}$ is remarkable because the physical model was calibrated without phase information for Fig. 3e and without information about one of the transmission coefficients for Fig. S2a.

In addition to this wavefront-shaping based wave control for coherent focusing, we also consider the possibility of controlling both the input wavefront and the metasurface configuration to maximize the deposited energy. To that end, we use the physical model to select a metasurface configuration out of the $10^5$ configurations from the training data set that we expect to yield the largest deposited energy. Even though these configurations were part of the calibration data, because the training data was phaseless in the case of Fig. 3e and lacked one relevant transmission coefficient in the case of Fig. S2a, our ability to identify a metasurface configuration that maximizes the deposited energy is remarkable. Our dictionary-search-based inverse-design algorithm consists in (i) establishing the dictionary by running the physics-based forward model for the $10^5$ configurations, (ii) evaluating the cost function (focused intensity) for each dictionary entry, and (iii) selecting the dictionary entry with the best cost function. By realizing that the interaction matrices for the $10^5$ configurations only differ regarding parts of their diagonal, we can evaluate the corresponding scattering matrices by updating a previous interaction matrix inverse using the Woodbury identity, as opposed to evaluating the interaction matrix inverse from scratch (see Sec. IV.A in ref. 38). For our Matlab implementation of this inverse-design algorithm, the CPU times are 25.1 s, 0.7 s and 0.002 s for these three steps on a laptop with an AMD Ryzen 7 PRO 4750U processor and 64 GB RAM.

**Coherent absorption.** We consider the problem of maximizing the wave energy absorbed in our scattering system by injecting a coherent monochromatic wavefront $\boldsymbol{\psi}_{\mathrm{in}}$ at 5.2 GHz through all four antennas. The main absorption mechanism of our system is Ohmic loss on the metallic walls; the absorption is hence distributed rather than spatially localized. The metasurface is in a known configuration $\mathbf{c}$ that is chosen randomly and has not been part of the calibration data set. $\boldsymbol{\psi}_{\mathrm{in}}$ is normalized so that its 2-norm is unity. Let the eigenvalues of the Hermitian matrix $\mathbf{S}^\dagger\mathbf{S}$ be denoted by $s_n$, where $0 \leq s_1 \leq s_2 \leq s_3 \leq s_4 \leq 1$, and the corresponding eigenvectors are $\boldsymbol{v}_n$. The reflected wave energy exiting the scattering system is $R = |\mathbf{S}\boldsymbol{\psi}_{\mathrm{in}}|^2$ and the absorbed energy is $1 - R$. The provably optimal input wavefront for coherently enhanced absorption is $\boldsymbol{\psi}_{\mathrm{in}}^{\mathrm{opt}} = \boldsymbol{v}_1$, i.e., the eigenvector of $\mathbf{S}^\dagger\mathbf{S}$ that is associated with the eigenvalue of smallest magnitude[45]. This constitutes our

benchmark (blue) in Fig. 3f and Fig. S2b. For physical-model-based wave control, we do not have access to **S** but only to our model's prediction $\mathbf{S}_{\mathrm{mod}}(\mathbf{c})$ thereof with associated eigenvectors $\boldsymbol{\upsilon}_n^{\mathrm{mod}}$ such that we use $\boldsymbol{\psi}_{\mathrm{in}}^{\mathrm{mod}} = \boldsymbol{\upsilon}_1^{\mathrm{mod}}$ as coherent input wavefront (red). To illustrate the importance of wavefront shaping, we also show a benchmark with a uniform wavefront $\boldsymbol{\psi}_{\mathrm{in}}^{\mathrm{uni}} = [111]/\sqrt{3}$ (yellow). Figure 3f and Fig. S2b show the reflected energy upon injecting $\boldsymbol{\psi}_{\mathrm{in}}^{\mathrm{opt}}$ (blue), $\boldsymbol{\psi}_{\mathrm{in}}^{\mathrm{mod}}$ (red) and $\boldsymbol{\psi}_{\mathrm{in}}^{\mathrm{uni}}$ (yellow) for 50 random and previously unseen metasurface configurations. Note that the fact that the reflected (and hence also the absorbed) energy is almost identical for $\boldsymbol{\psi}_{\mathrm{in}}^{\mathrm{opt}}$ and $\boldsymbol{\psi}_{\mathrm{in}}^{\mathrm{mod}}$ is remarkable because the physical model was calibrated without phase information for Fig. 3f and without information about one of the transmission coefficients for Fig. S2b.

In addition to this wavefront-shaping based wave control for coherently enhanced absorption, we also consider the possibility of controlling both the input wavefront and the metasurface configuration to maximize the absorbed energy. To that end, we use the physical model to select a metasurface configuration out of the $10^5$ configurations from the calibration data set that we expect to yield maximal absorption of energy. Even though these configurations were part of the calibration data, because the latter was phaseless in the case of Fig. 3f and lacked one relevant transmission coefficient in the case of Fig. S2b, our ability to identify a metasurface configuration that maximizes the coherently enhanced absorbed energy is remarkable. Details for the dictionary-search-based inverse-design algorithm are similar to those described for coherent focusing above.

**QPSK backscatter communications.** We consider the problem in which Alice seeks to transfer information to Bob by encoding her message into already existing ambient waves via the metasurface configuration using quadrature-phase-shift-keying (QPSK). For simplicity, we chose the antenna with index $i$ as the source of the ambient continuous waves at 5.2 GHz which are emitted with constant amplitude (i.e., without any modulation), and we choose the antenna with index $j$ as the intended receiver (Bob). To transmit information to Bob, in backscatter communications Alice controls the metasurface configuration $\mathbf{c}$ (rather than modulating the signal emitted by antenna $i$). Because some paths from antenna $i$ (transmitter) to antenna $j$ (receiver) are in general not impacted by the metasurface, $\langle S_{ji}(\mathbf{c})\rangle_{\mathbf{c}} \neq 0$, Bob considers $S_{ji} - \langle S_{ji}\rangle$ to decode Alice's message. Specifically, to perform QPSK backscatter communications, Alice switches between four carefully chosen metasurface configurations $\mathbf{c}_{00}$, $\mathbf{c}_{01}$, $\mathbf{c}_{11}$, and $\mathbf{c}_{10}$ such that $S_{ji} - \langle S_{ji}\rangle$ has the same amplitude for all four configurations, but the phase of $S_{ji} - \langle S_{ji}\rangle$ takes the values of $e^{i\theta}$, $e^{i(\theta+\frac{\pi}{2})}$, $e^{i(\theta+\pi)}$, or $e^{i(\theta+\frac{3\pi}{2})}$ if Alice wants to send '00', '01', '11', or '10', respectively[47]. Here, $\theta$ is a global phase constant without physical meaning. For ease of visualization, $\theta$ is set to zero to display the constellation diagrams in Fig. 3g and Fig. S2c. As shown in ref. 47, with these four metasurface configurations it is also possible to implement QPSK massive backscatter communications if the emitter radiates modulated WiFi waves rather than continuous waves.

Using our physical model, the four metasurface configurations $\mathbf{c}_{00}$, $\mathbf{c}_{01}$, $\mathbf{c}_{10}$, and $\mathbf{c}_{11}$ are chosen out of the $10^5$ configurations from the training data set. Our ability to identify four suitable metasurface configurations for backscatter QPSK is remarkable in the case of Fig. 3g because the calibration data was phaseless, and in the case of Fig. S2c because the utilized transmission coefficient $S_{ji}$ was not included in the calibration data. Details for the dictionary-search-based inverse-design algorithm are similar to those described for coherent focusing above.

## Data availability
All data needed to evaluate the conclusions in the paper are present in the paper and/or the Supplementary Materials.

## Code availability
The codes underlying the presented work are available from the corresponding author upon request.

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

## Acknowledgements

P.d.H. acknowledges stimulating discussions with Y.C. Eldar and O.D. Miller. The metasurface prototype was purchased from Greenerwave. This work was supported by: CNRS prématuration program through the project MetaFilt (P.d.H); European Union's Regional Development Fund; French region of Brittany and Rennes Métropole through the contrats de plan État-Région program (project SOPHIE/STIC & Ondes).

## Author contributions

P.d.H. conceived the project and wrote the code that calibrates the physical model. J.S. and P.d.H. conducted experimental work. H.P. and P.d.H. conducted numerical work. L.L.M. and P.d.H. significantly contributed to the project with thorough discussions regarding execution and interpretation. P.d.H. wrote the manuscript.

## Competing interests

The authors declare no competing interests.
