## [Peer Review File · Nature Communications]

Experimentally realized physical-model-based frugal wave control in metasurface-programmable complex mediaEditorial Note: This manuscript has been previously reviewed at another journal that is not operating a transparent peer review scheme. This document only contains reviewer comments and rebuttal letters for versions considered at *Nature Communications*.

REVIEWERS' COMMENTS

Reviewer #2 (Remarks to the Author):

The authors have effectively addressed all my comments and provided clear explanations regarding the significant contributions of the manuscript in comparison to existing works. Overall, the paper demonstrates novelty and technical soundness. Therefore, I recommend its publication in Nature Communications. No additional comments are necessary.

Reviewer #3 (Remarks to the Author):

This manuscript has been transferred to Nature Communications after three referees have commented on its publishability for Nature.

In their rebuttal letter, the authors provide substantial input on the referees' questions and have updated the manuscript accordingly. I appreciate the detailed replies and the amendments to the article, which has now clearly improved from my perspective. In particular, I do think that the authors have now managed to explain better what the strengths and novelty of their work really are, which will clearly help to embed these advances into the state-of-the-art in the literature.

As mentioned already in my first report, I do think that this manuscript fits well into Nature Communications and with the implemented changes I can now fully recommend its publication without any further changes.

Reviewer #2 (Remarks to the Author):

The authors have effectively addressed all my comments and provided clear explanations regarding the significant contributions of the manuscript in comparison to existing works. Overall, the paper demonstrates novelty and technical soundness. Therefore, I recommend its publication in Nature Communications. No additional comments are necessary.

→ No further action required.

Reviewer #3 (Remarks to the Author):

This manuscript has been transferred to Nature Communications after three referees have commented on its publishability for Nature.

In their rebuttal letter, the authors provide substantial input on the referees' questions and have updated the manuscript accordingly. I appreciate the detailed replies and the amendments to the article, which has now clearly improved from my perspective. In particular, I do think that the authors have now managed to explain better what the strengths and novelty of their work really are, which will clearly help to embed these advances into the state-of-the-art in the literature.

As mentioned already in my first report, I do think that this manuscript fits well into Nature Communications and with the implemented changes I can now fully recommend its publication without any further changes.

→ No further action required.